# Anti-resonant acoustic waveguides enabled tailorable Brillouin scattering on chip

Peng Lei[1], Mingyu Xu[1], Yunhui Bai[1], Zhangyuan Chen[1] & Xiaopeng Xie [1]✉

Empowering independent control of optical and acoustic modes and enhancing the photon-phonon interaction, integrated photonics boosts the advancements of on-chip stimulated Brillouin scattering (SBS). However, achieving acoustic waveguides with low loss, tailorability, and easy fabrication remains a challenge. Here, inspired by the optical anti-resonance in hollow-core fibers and acoustic anti-resonance in cylindrical waveguides, we propose suspended anti-resonant acoustic waveguides (SARAWs) with superior confinement and high selectivity of acoustic modes, supporting both forward and backward SBS on chip. Furthermore, this structure streamlines the design and fabrication processes. Leveraging the advantages of SARAWs, we showcase a series of breakthroughs for SBS within a compact footprint on the silicon-on-insulator platform. For forward SBS, a centimeter-scale SARAW supports a large net gain exceeding 6.4 dB. For backward SBS, we observe an unprecedented Brillouin frequency shift of 27.6 GHz and a mechanical quality factor of up to 1960 in silicon waveguides. This paradigm of acoustic waveguide propels SBS into a new era, unlocking new opportunities in the fields of optomechanics, phononic circuits, and hybrid quantum systems.

Brillouin nonlinearities arise from the coupling of photons and phonons[1–4]. Due to its unique characteristics derived from acoustic phonons, SBS has found widespread applications in the realms of signal generation and processing[5–18], such as narrow linewidth laser[16,17], microwave photonic filters[5–8], slow and fast light[9–11], distributed sensing[13–15], and optical non-reciprocal devices[18]. Recently, integrated photonics has enabled better confinement and manipulation of optical and acoustic modes[19–27], ushering in a new era of photon-phonon interaction.

While optical mode confinement methods are well established, how to confine acoustic modes is still evolving, which now primarily relies on three strategies. The first approach is acoustic total internal reflection, which is generally applied in materials with low stiffness[19,20], such as chalcogenides. Nonetheless, these materials are difficult to integrate with standard silicon photonic circuits. The second solution, based on acoustic impedance, involves isolating waveguides from the substrate to effectively prevent acoustic leakage[21,22]. Whereas, this approach has difficulty in flexibly selecting acoustic frequencies, and

necessitates complex fabrication processes. The third strategy utilizes phononic bandgaps to obstruct the spread of acoustic waves[28–31]. However, its complex structures pose difficulties for design and manufacturing and result in low robustness.

Emerging anti-resonant reflection offers an alternative approach to achieve field confinement[32–36]. This approach has been successfully applied in hollow-core fibers[32–34], effectively confining optical modes within the lower-refractive-index air core. Compared with photonic bandgap hollow-core fibers[32], anti-resonant hollow-core fibers have a simpler structure and lower loss. Anti-resonant reflection has been theoretically proposed to confine acoustic modes of both forward and backward SBS in cylindrical waveguides[35]. However, they are not compatible with integrated photonic platforms.

In this paper, we transfer the concept of anti-resonant reflection to integrated platforms and demonstrate a novel suspended anti-resonant acoustic waveguide (SARAW), proving its effectiveness in the confinement of phonons. By designing the anti-resonant structures, we can flexibly control the acoustic modes without affecting optical

[1]State Key Laboratory of Advanced Optical Communication Systems and Networks, School of Electronics, Peking University, 100871 Beijing, China.
✉e-mail: xiaopeng.xie@pku.edu.cn

modes. This characteristic enables us to manipulate the acoustic mode distribution, select the eigenfrequency, and optimize the mechanical quality factor $Q_m$. Consequently, SARAWs can support both forward and backward SBS and achieve a series of extraordinary results of SBS on the silicon-on-insulator (SOI) platform. For forward SBS, we attain a Brillouin gain coefficient $G_B$ of 3530 W$^{-1}$m$^{-1}$, over 2000 times larger than that of standard single-mode fibers. The threshold for achieving Brillouin net gain is remarkably low (<5 mW). Under a modest pump power, a large net gain of up to 6.4 dB can be realized with a compact footprint. These results highlight that SARAWs can significantly enhance the coupling strength of optical and acoustic waves. For backward SBS, we observe a record-high Brillouin frequency shift of 27.6 GHz and an unprecedented $Q_m$ of 1960 in integrated silicon waveguides. These breakthroughs show that acoustic waves in SAR-AWs can operate at frequencies extending to millimeter-wave bands, and exhibit extremely low propagation loss. In terms of fabrication and design, SARAWs eliminate the need for overlay exposure in the

fabrication processes, thus suppressing the harmful inhomogeneous broadening of Brillouin resonance. It also allows for a simplified waveguide design process utilizing genetic algorithms[37,38].

## Results

### Design and fabrication of SARAWs

We first introduce the principle of acoustic anti-resonant reflection with the planar waveguide, as depicted at the top of Fig. 1a. To confine the acoustic field within the middle layer, the side layers with slower acoustic velocities are incorporated as anti-resonant reflecting layers, which can be conceptualized as Fabry-Pérot cavities. By manipulating cavities thicknesses, the anti-resonance states can be achieved. Consequently, the acoustic wave is unable to traverse the side layers and is reflected to the middle layer, leading to the acoustic field distribution outlined by the black line in Fig. 1a. Generally, different velocity layers are composed of distinct materials[32,35], such as silicon and silica. However, implementing this transverse multilayer structure on the

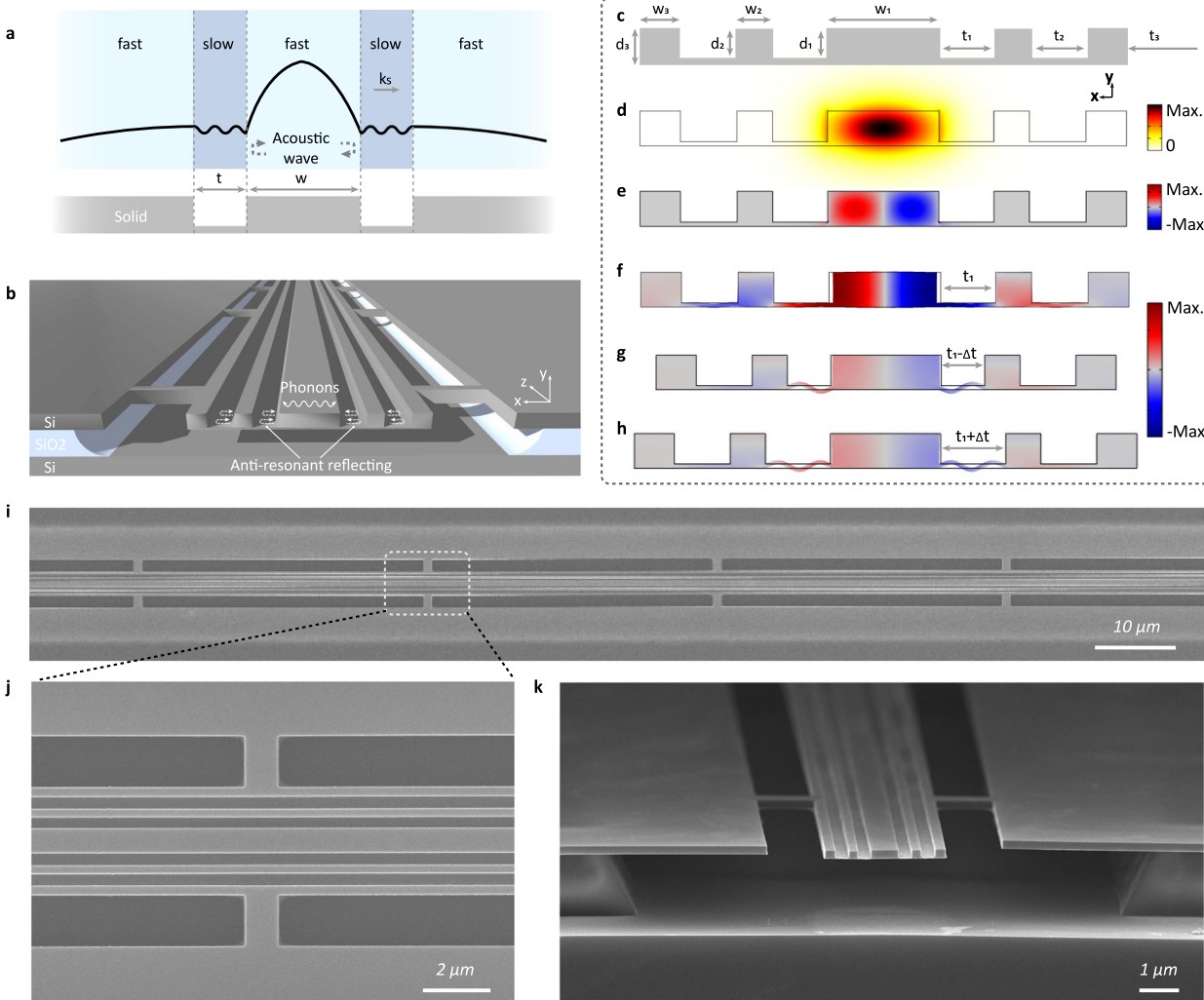

**Fig. 1 | Suspended anti-resonant acoustic waveguide on the SOI platform. a** The principle of anti-resonant reflection and its implementation on chip. **b** Schematic of the SARAW. **c** Cross-section of the SARAW and its geometric parameters. These parameters are axisymmetric. $W_i$ (i = 1, 2, 3) corresponds to the width of the unetched region, $t_i$ corresponds to the width of the etched slot, and $d_i$ corresponds to the etching depth of the etched slot. Benefiting from the loading-effect etching process, wherein a wider $t_i$ corresponds to a deeper $d_i$, we can control the etching depth $d_i$ by adjusting the slot width $t_i$. $t_3$ is set to sufficiently large to achieve overetching, which enables the hydrofluoric acid solution to remove the silica substrate. Consequently, the waveguides are fully suspended in air. **d** The electric

field of the optical mode. **e** The x component of the electrostrictive force. **f** The elastic displacement field of the optimized acoustic mode. Here, slot width $t_1$ meets the anti-resonance condition. Figure (**e**, **f**) use the optimized parameters for the forward SBS with a waveguide width $W_1$ of 700 nm. The detailed geometric parameters are shown in Supplementary Section III. **g, h** The elastic displacement field when acoustic mode under the resonance condition. In Fig. (**f**–**h**), the color bar corresponds to the magnitude of the x component of the elastic displacement field. The deformations depict the cross-sectional displacement fields. **i, j** Top-down SEM image of the fabricated SARAW (**i**) and a magnified view (**j**). **k** Cross-sectional SEM image of the fabricated SARAW. Scale bars, 10 μm (**i**); 2 μm (**j**); and 1 μm (**k**).

chip is challenging. Here, we propose a new approach to address this issue. Considering the situation in the suspended solid membrane, as illustrated at the bottom of Fig. 1a, slots with width t are etched onto both sides of the central waveguide. In the slot regions, the boundary between air and solid geometrically softens the structural response of solid film[3], thus lowering its effective acoustic velocity and achieving anti-resonant reflecting layers.

Following this approach, we propose the suspended anti-resonant acoustic waveguide as shown in Fig. 1b, c. The device consists of a central waveguide flanked by two sets of etched slots, and the entire suspended structure is supported by a series of tethers. Since silicon exhibits excellent optical and acoustic properties, such as a high refractive index and a low acoustic loss, and is compatible with complementary metal-oxide-semiconductor (CMOS) technology, it becomes a conducive medium for robust photon-phonon interactions. We deploy SARAWs on the SOI platform and demonstrate its improvement and tailorability on Brillouin nonlinearities.

We first investigate the properties of the SARAWs using forward SBS, which has a larger gain coefficient $G_B$ than backward SBS. Through the meticulous adjustment of anti-resonant structures using a genetic algorithm[37,38] (see "Methods"), SARAWs can be optimized to achieve the largest $G_B$. We showcase the SARAW with a central waveguide width $W_1$ of 700 nm, see Supplementary Section III for detailed parameters. The SARAW supports a fundamental transverse electric-like optical mode (Fig. 1d), and the corresponding electrostrictive force is depicted in Fig. 1e. The acoustic mode of SARAWs, compared to the rib waveguides[22], is squeezed towards the central waveguide, as shown in Fig. 1f, indicating the effectiveness of anti-resonant reflection in acoustic mode confinement. Furthermore, the similar distribution of the electrostrictive force and acoustic mode suggests a large photon-phonon overlap, closing to that of the fully suspended rectangular waveguide. To further demonstrate the effect of anti-resonant reflection, we introduce a variation ($t/t_1 = 20\%$) to the slot width ($t_1$), making SARAWs to operate in resonance conditions. As shown in Fig. 1g, h, the acoustic intensity at the central waveguide decreases sharply, and a significant portion of the acoustic energy leakages into the slot region, inducing a significant increase in the deformation. This results in a reduction in photon-phonon overlap. Hence, by adjusting the geometric parameters, SARAWs can be flexibly switched between the anti-resonance and resonance states, effectively manipulating the acoustic confinement and the photon-phonon coupling strength. Moreover, this manipulation does not influence the optical mode and can be performed independently.

SARAWs also streamline the fabrication processes and play a significant role in reducing the inhomogeneous broadening of Brillouin resonance. Previous suspended SBS waveguides typically necessitate overlay exposure[22,29], involving at least two exposure steps and two etching steps, and all steps call for exceptional alignment precision. Furthermore, the alignment mismatches in the overlay exposure process would introduce inhomogeneous broadening of Brillouin resonance, exacerbating SBS performance. Particularly, the degradation of alignment mismatches is significantly magnified in spiral waveguides (see Supplementary Section VIII). Here, for the first time, we innovatively propose an etching technique based on the loading effect to fabricate the whole structure via a single exposure and etching step (see "Methods" and Supplementary Section IV). The loading effect[39], where the etch rate depends on pattern width in etching processes, generally acts as a barrier to achieving depth uniformity. Yet, this property provides the opportunity to attain different etching depths in a single etching step. Given the distinctive structure of the SARAW, which requires different slots depths ($d_1 \sim d_3$), the loading-effect etching technique can be effectively applied, avoiding the costs and yield losses caused by multiple fabrication cycles. By excluding the overlay exposure requirement, the loading-effect etching technique can accommodate high fabrication precision (<5 nm),

and mitigate the inhomogeneous broadening. By introducing rectangle spiral waveguides (see Supplementary Section VIII), we can achieve centimeter-scale-long waveguides in a compact footprint with improved consistency, yielding higher $Q_m$ and $G_B$ compared to straight waveguides. On top of these design and fabrication processes, the SARAWs are analyzed by scanning electron microscopy (SEM) (Fig. 1i–k), which demonstrate excellent structural stability and uniformity. Benefiting from the streamlined fabrication process and compatibility with CMOS technology, these results exhibit great reproducibility and scalability, facilitating the large-scale integration of Brillouin-based devices.

## Demonstration of anti-resonance with SARAWs

Initially, we validate the tailorability of SARAWs through simulation in terms of the acoustic frequency and mode distribution. For the anti-resonant structure, the slot width $t_1$ can effectively control the anti-resonance state and influence the acousto-optic interaction strength. Therefore, we varied the slot width $t_1$ of the SARAW from 200 to 400 nm, with the other parameters kept the same as in Fig. 1f. Through the simulation (see "Methods" and Supplementary Section III), we tracked the acoustic frequency and coupling factor $G_B/Q_m$, which reflects the photon-phonon overlap. The simulation results, shown in Fig. 2b, indicate a decrease in the frequency of the Brillouin-active acoustic mode from 6.8 GHz to 5.9 GHz as the width $t_1$ increases. This trend results from the influence of another acoustic mode (see Supplementary Section III). During the sweep, three resonance conditions are met ($t_1 = 225, 325,$ and $388$ nm), with two anti-resonance conditions in between. When slot width $t_1$ meets the resonance condition, avoided mode crossing[40] is observed. This phenomenon arises from the strong coupling between the acoustic mode in the central waveguide and the shear modes in the slot regions, leading to a significant decrease in photon-phonon overlap. The uneven appearance of resonance conditions when $t_1$ increases is mainly attributed to the consideration of the loading effect in the simulations (see Supplementary Section III).

Using the parameters in the simulation, we fabricated a series of 5-mm-long SARAWs and conducted the heterodyne four-wave mixing (FWM) experiment[22,29]. This experimental setup, shown in Fig. 2a, provides excellent sensitivity and avoids the impact of coupling loss fluctuations, thus ensuring robust and reliable results. We extract Brillouin frequency shift, $G_B$, and $Q_m$ by fitting the obtained asymmetric Fano-like spectral lines (see Supplementary Section V). In order to clearly compare the simulation and experimental results, acoustic frequency and coupling factor $G_B/Q_m$ are depicted in Fig. 2c, d respectively. Avoided mode crossing and decrease of coupling factor $G_B/Q_m$ under resonance conditions are observed, showing good agreement with the simulation. As shown in Fig. 2e, the dips and peaks of mechanical quality factor $Q_m$ illustrate the noticeable difference between resonance and anti-resonance conditions for the acoustic confinement effect. The difference is further illustrated by Brillouin spectra obtained by the heterodyne FWM experiment under corresponding conditions (Fig. 2f, g). Under the anti-resonance condition (Fig. 2f), the fitted $G_B$ and $Q_m$ reach up to 3530 W⁻¹m⁻¹ and 680, indicating that the acoustic mode is effectively confined within the central waveguide. While under the resonance condition (Fig. 2g), both $G_B$ and $Q_m$ decrease dramatically by over 80%, reaching 540 W⁻¹m⁻¹ and 120, respectively, as the acoustic mode energy disperses into the slot regions. The above simulation and experimental results solidly validate the effectiveness of SARAWs for acoustic confinement and manipulation. Compared to suspended rectangular waveguides without anti-resonant structures[41], SARAWs can compress the acoustic mode into the central waveguide, thereby mitigating the impact of sidewall roughness on the $Q_m$, resulting in a larger $G_B$. Moreover, the presence of the anti-resonant structure can isolate the optical mode from the supporting tethers, reducing optical mode mismatch loss. These

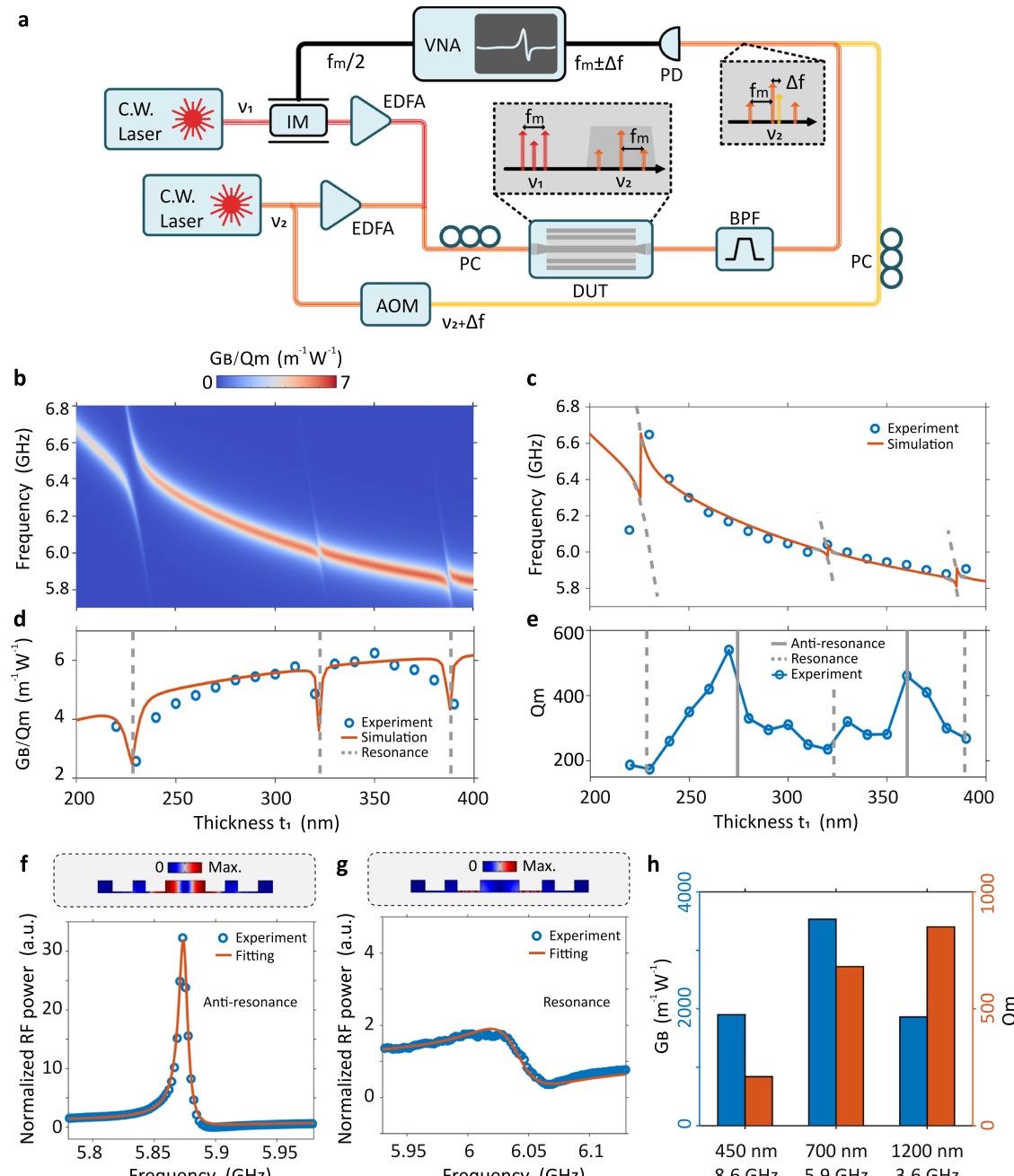

**Fig. 2 | The demonstration of anti-resonance and experimental results.**
**a** Diagram of the heterodyne FWM experiment. Two continuous-wave (C.W.) lasers operate at frequencies $v_1$ and $v_2$ (corresponding to wavelengths of 1550 and 1552 nm). The upper laser generates sidebands ($v_1 \pm f_m/2$) via an intensity modulator (IM) with the modulation signal ($f_m/2$) provided by a vector network analyzer (VNA). After amplified by an erbium-doped fiber amplifier (EDFA), these sidebands serve as the pump light for the device under test (DUT). The second C.W. laser is split into two paths by a coupler. The upper branch is amplified by an EDFA as the probe light. The lower branch undergoes a frequency shift of $\Delta f$ via an acousto-optic modulator (AOM) as a reference signal. Within the DUT, the pump light ($v_1 \pm f_m/2$) and the probe light ($v_2$) generate two sidebands at $v_2 \pm f_m$ through FWM (the gray area beneath the DUT). To measure the intensity variations of the FWM-generated sidebands when $f_m$ scans over the Brillouin frequency shift, $v_2 \pm f_m$ are filtered out via a cavity band-pass filter (BPF) (the dark gray trapezoidal box on the right below

the DUT), and then mixed with $v_2 + \Delta f$ on a photodetector (PD), yielding signals at $f_m \pm \Delta f$. The corresponding FWM variation curves can be obtained on the VNA, corresponding to Stokes ($f_m + \Delta f$) and anti-Stokes ($f_m - \Delta f$) components. **b** The influence of resonance/anti-resonance condition on Brillouin frequency shift and coupling factor $G_B/Q_m$ with different slot width ($t_1$). To facilitate observation, the linewidth of the acoustic modes is set to 50 kHz. **c–e** Experimental and simulated results show the impact of anti-resonant reflection on Brillouin frequency shift, $G_B/Q_m$, and $Q_m$, respectively. The elastic displacement magnitude (top) and measurement results (bottom) of the heterodyne FWM experiment (anti-Stokes sideband, $v_2 + f_m$) in SARAWs under the anti-resonance (**f**) and resonance (**g**) condition. The waveguide width $W_1 = 700$ nm. The a.u. denotes arbitrary units. **h** The experimental attained $G_B$ and $Q_m$ of optimized SARAWs for $W_1 = 450$, 700, and 1200 nm, with the acoustic eigenfrequency of 8.6, 5.9, and 3.6 GHz, respectively.

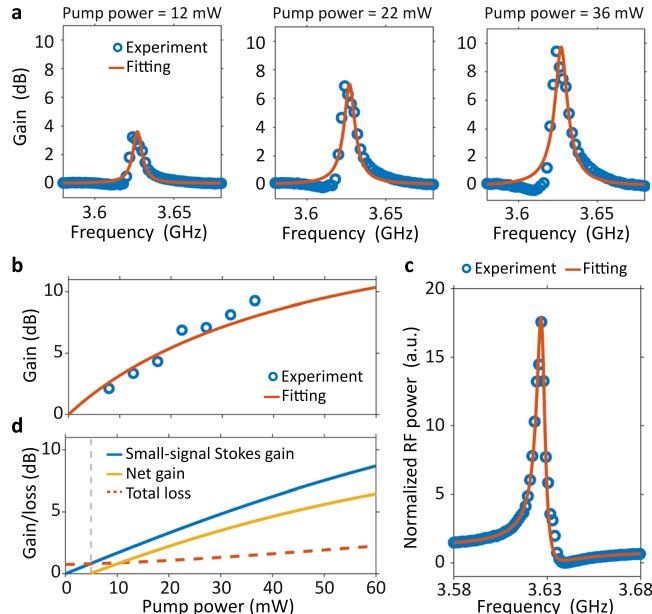

**Fig. 3 | Measurement of Brillouin gain. a** Normalized Stokes transmitted spectra for pump powers of 12, 22, and 36 mW. **b** The experimental and simulation results of three-tone direct gain. **c** The heterodyne FWM experiment result of the same SARAW. **d** Small-signal Stokes gain and net gain. The threshold for net gain is 5 mW.

attributes underscore the comprehensive superiority and inherent flexibility of SARAWs.

Utilizing the manipulating ability and frequency selectivity of anti-resonance, we optimize and fabricate 5-mm-long SARAWs with different central waveguide widths ($W_1$ = 450, 700, and 1200 nm) to obtain the respective maximum $G_B$. The experimentally attained Brillouin frequency shift, $G_B$, and $Q_m$ are shown in Fig. 2h. As the waveguide width increases, the waveguide acquires a larger mode field area and becomes less sensitive to sidewall roughness. Therefore, the eigenfrequency and coupling factor $G_B/Q_m$ decrease gradually, while $Q_m$ increases accordingly. The 450-nm SARAW exhibits the highest $G_B/Q_m$ of 8.57, but suffers more from the inhomogeneous broadening caused by sidewall roughness, resulting in a low $Q_m$. The 700-nm SARAW achieve a remarkable $G_B$ of 3530 W$^{-1}$m$^{-1}$ in silicon waveguides with a balance of $G_B/Q_m$ and $Q_m$. And the 1200-nm SARAW shows a higher $Q_m$ of up to 850. Consequently, SARAWs can be flexibly extended to different waveguide widths, allowing for the attainment of distinct acoustic frequencies, $G_B$ and $Q_m$.

## Large Brillouin net gain in SARAWs
The Brillouin net gain, which takes the Brillouin gain coefficient $G_B$ and optical loss into account, is typically a comprehensive figure of merit for Brillouin-based applications. Hence, we conducted further design and experiment to illustrate the potential of SARAWs in achieving large Brillouin net gain. Benefiting from the properties of low optical loss and minimal inhomogeneous broadening, SARAWs with a central waveguide width of 1200 nm are suitable for attaining large net gain. We examine Brillouin interactions in a 2.5-cm-long rectangular spiral SARAW with a compact footprint of 1900 μm × 460 μm.

We carried out a three-tone direct gain experiment[29] (see Supplementary Section VI) to obtain the net gain, owing to its simpler setup compared to traditional small-signal gain experiment. Figure 3a shows three Brillouin gain spectra measured for pump powers of 12, 22, and 36 mW. These gain spectra, which illustrate the relative change in the probe intensity, reveal a Brillouin resonance with a high mechanical quality factor ($Q_m$ = 660, see Supplementary Section VI for a detailed fitting process) at 3.63 GHz. The probe intensity is

significantly amplified by approximately 10 dB under 36 mW pump power. There is a slight discrepancy between the experimental results and the fittings, which could be attributed to the phase drift of the modulator and the acoustic eigenfrequency drift. Figure 3b shows the Stokes gain at Brillouin resonance as a function of pump power. To satisfy the small signal approximation, the pump power is kept below 40 mW. By numerically solving the three-tone coupling equations (see Supplementary Section VI), the fitted $G_B$ is determined as 1700 W$^{-1}$m$^{-1}$ (Fig. 3b). It is of great consistency with the independent measurement of the same waveguide performed through heterodyne FWM measurement ($G_B$ = 1670 W$^{-1}$m$^{-1}$, $Q_m$ = 650), as is shown in Fig. 3c. Compared to the 5-mm-long SARAW with the same structure parameters (Fig. 2h, central waveguide width $W_1$ = 1200 nm), Brillouin gain coefficient $G_B$ and mechanical quality factor $Q_m$ here are slightly reduced due to the inhomogeneous broadening under long waveguide length. Based on the results from the above experiments (Fig. 3b, c), the small-signal Stokes gain and the net gain are numerically calculated (see Supplementary Section VI) and shown in Fig. 3d. Benefiting from the low optical loss (0.3 dB/cm, see Supplementary Section I) and large Brillouin gain, the threshold for net gain is below 5 mW. Considering the maximum on-chip pump power of 57 mW constrained by EDFA and coupling loss (single-sideband 5.4 dB), SARAWs can support a net gain of up to 6.4 dB and an on-off gain of 8.7 dB. These results mark the present apex of Brillouin gain levels in silicon waveguides and can be further improved with higher on-chip power and longer waveguide length. However, it should be noted that, on the SOI platform, further increases in pump power may lead to notable nonlinear absorption losses (see Supplementary Section I). This factor would become the primary constraint for achieving larger Brillouin net gain. The introduction of pump light in far-infrared wavelengths or the free-carrier suppression techniques[42] could potentially overcome this limitation.

## Backward SBS in SARAWs
Backward SBS, mediated by the interaction of two counter-propagating optical waves with a phase-matched longitudinal acoustic wave, exhibits distinct characteristics compared to forward SBS. Acoustic modes in backward SBS exhibit higher acoustic frequency and quality factor. The longitudinal wave nature of the backward SBS acoustic mode necessitates an extensive waveguide length to build up to its full strength. However, existing suspended waveguides inevitably introduce periodic supporting structures[22,43], leading to the dissipation of acoustic waves. Meanwhile, the material property of the photo-elastic coefficient makes it challenging to observe backward SBS in silicon waveguides[2]. The maximum on-off gain, $G_B$, and Brillouin frequency shift for backward SBS in silicon waveguides achieved so far are only 0.2 dB, 359 W$^{-1}$m$^{-1}$, and 13.7 GHz[21], respectively. However, in SARAWs, the anti-resonant structures separate the acoustic wave from the periodic supporting tethers. This separation enables the acoustic wave to build up to its full strength, thereby facilitating robust support for backward SBS.

We designed and fabricated 2-cm-long SARAWs with central waveguide widths $W_1$ = 450, 700, and 1200 nm. Utilizing the setup in Supplementary Section VII, we observed a bell-shaped acoustic mode under all three waveguide widths (Fig. 4a–c). Same as the trends in forward SBS, as the central waveguide width $W_1$ increases, the photon-phonon overlap decreases and the mechanical quality factor increases. With a balance of $G_B/Q_m$ and $Q_m$, the 700 nm SARAW achieves a backward Brillouin gain coefficient $G_B$ of 600 W$^{-1}$m$^{-1}$ (Fig. 4b), nearly doubling the current record observed in silicon waveguides[21]. And surprisingly, the 1200-nm SARAW (Fig. 4c) attains an unprecedented $Q_m$ of up to 1960 in integrated waveguides, significantly exceeding that of forward SBS acoustic modes. Furthermore, a record-high acoustic frequency of 27.6 GHz is observed in another acoustic mode (Fig. 4d) utilizing a 450-nm SARAW. The geometric parameters used here have been meticulously optimized for this specific acoustic mode,

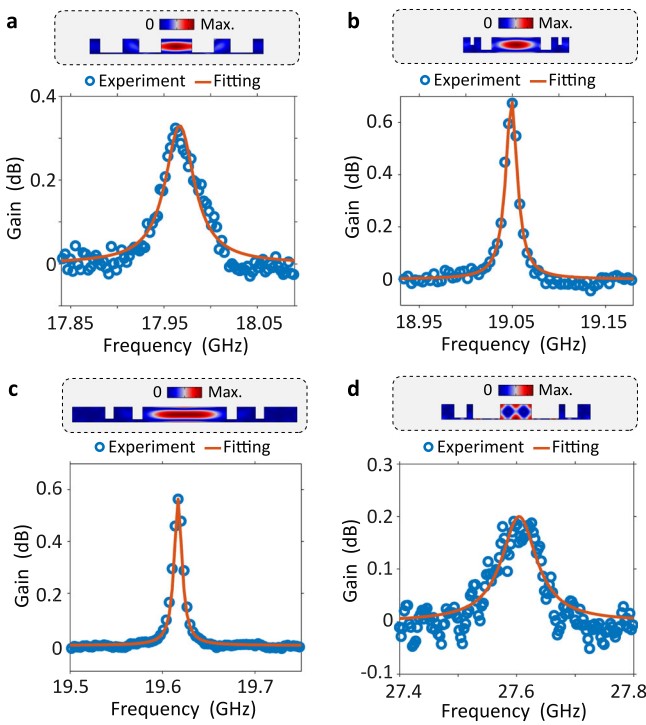

**Fig. 4 | Measurement of Backward SBS. a–d** The elastic displacement magnitude of the corresponding acoustic mode field (upper gray region) and backward SBS gain spectra (lower). Figures (**a**–**c**) correspond to the optimized acoustic modes with central waveguide width $W_1$ = 450, 700, and 1200 nm, using the genetic algorithm. Figure (**d**) illustrates the outcomes obtained through optimization for higher-order acoustic modes within a 450-nm SARAW. The values of Brillouin gain coefficient $G_B$ (W$^{-1}$m$^{-1}$) and mechanical quality factor $Q_m$ are: (530, 470); (600, 1300); (430, 1960); (480, 380), corresponding to figure (**a**–**d**), respectively.

mechanical quality factor $Q_m$, and Brillouin gain coefficient $G_B$. Meanwhile, the innovative loading-effect-based etching technique minimizes the manufacturing cost while maintaining excellent fabrication precision. It lessens the degradation of Brillouin resonance caused by inhomogeneous broadening while minimizing the overall footprint. Furthermore, this scheme is portable to other material platforms, rendering it highly suitable for integrated acousto-optic devices.

In our work, the net gain in SARAWs is mainly constrained by the coupling loss and the roughness of the waveguide sidewall, which are limited by our fabrication capabilities. These limitations hinder further increases in on-chip pump power and mechanical quality factor $Q_m$, and impede further reductions in optical propagation loss. However, referring to previous work[21], the utilization of CMOS pilot line in the foundry could potentially yield lower coupling loss, higher $Q_m$, and lower optical loss. These advancements promise a performance enhancement of more than twofold for current net gain results.

We would like to emphasize that SARAWs exhibit exceptional characteristics in backward SBS. The backward SBS acoustic mode exhibits a significantly larger group velocity and an extended phonon lifetime than the forward SBS, resulting in a large acoustic decay length. This is also a crucial characteristic for the acoustic waveguides, making SARAWs a promising candidate for supporting the burgeoning field of phononic circuits[45]. Furthermore, the promotions in acoustic mode eigenfrequency and $Q_m$ underscore the potential of SARAWs. A maximum $Q_m$ of 1960 and an unprecedented acoustic eigenfrequency of 27.6 GHz are particularly advantageous for microwave photonics and signal processing devices for 5G and future communication systems, including microwave generations[12] and acoustic wave filters[46]. In addition, the acoustic frequency of 19.6 GHz and $Q_m$ of 1960 indicate a product of frequency and quality factor $f \times Q_m$ exceeding $3.8 \times 10^{13}$ Hz, which is important for quantum systems. It is expected to be further improved in phonon lasers[47] based on SARAWs.

In summary, SARAWs incorporate innovations in acoustic anti-resonance theory, intelligent algorithm empowered design, and nanofabrication technique, offering a comprehensive solution for integrated photon-phonon interaction platforms. It heralds a new era for SBS, paving the way for breakthroughs in optomechanics[48], phononic circuits[45], and hybrid quantum systems[49].

## Methods
### Device fabrication
The SARAW was fabricated from the 220 nm silicon layer of a SOI wafer with the ⟨100⟩ crystal orientation utilizing electron-beam lithography (EBL) and inductively coupled plasma (ICP) etching. With loading-effect-based etching technique, SARAW can be fabricated with one exposure step and one etching step. Whereafter, the oxide undercladding was then released by immersion in 10% hydrofluoric acid. The SARAW employs a rectangular spiral for long waveguides. The 2.5-cm-long waveguide has a footprint of 1900 μm × 460 μm (see Supplementary Section IV).

### Genetic algorithm and finite element simulation
First, we developed the main program and invoked the Genetic Algorithm Toolbox in MATLAB. Then, we simulated acoustic and optical modes through the finite element solver COMSOL, calculating the total $G_B/Q_m$ produced by electrostrictive force and radiation pressure. Subsequently, the obtained $G_B/Q_m$ was further iterated by the genetic algorithm until the algorithm converged. It should be noted that due to the loading effect, slot width $t_i$ dictates the slot depth $d_i$ (Fig. 1c). Therefore, we measured the relationship curve between $t_i$ and $d_i$ caused by the loading effect through separate experiments, and then compensated it into the main program, ensuring that our optimization results were consistent with the actual fabricated waveguides (see Supplementary Section III).

distinguishing from the parameters used in Fig. 4a. It underscores the selectivity of SARAWs for specific acoustic modes and frequencies. However, this acoustic mode exhibits lower $G_B$ of 480 W$^{-1}$m$^{-1}$ and $Q_m$ of 380. It may be attributed to the fact that a significant portion of the acoustic mode is distributed near the waveguide boundaries, which results in a smaller acousto-optic overlap and makes it more vulnerable to surface roughness. Thus this mode has not been observed in SARAWs with central waveguide widths $W_1$ of 700 and 1200 nm.

The record-breaking results above demonstrate that SARAWs can flexibly enhance and manipulate acoustic modes for backward SBS. As a result, SARAWs support acoustic modes with high group velocity (>3000 m/s), and the corresponding acoustic decay length of over 50 μm, offering a low-loss and long-distance transmission platform for acoustic waves. However, the observation of backward SBS net gain remains hindered by the limitation imposed by the intrinsic photo-elastic coefficient of silicon. Promising avenues for overcoming this challenge include transplanting SARAWs onto a chalcogenide glass platform or selecting SOI wafers with diverse crystal orientations[44].

## Discussion
Leveraging genetic-algorithm-based optimization scheme and loading-effect-based fabrication technique, SARAWs can be rapidly designed and fabricated. For design scheme, the genetic algorithm eliminates the need for time-consuming parameter sweep and greatly simplifies the SARAW design process. Moreover, it allows for the rapid and specific optimization of various merit figures in SBS. This versatility is crucial for SBS-based applications with different requirements depending on the use-case. For instance, SBS-based microwave photonic filters[5–8] with different center frequencies, bandwidths, and out-of-band suppression ratios demand diverse Brillouin frequency shift,

## Data availability

The data that supports the plots within this paper and other findings of this study have been deposited in the Zenodo database (https://zenodo.org/doi/10.5281/zenodo.10946980). All other data used in this study are available from the corresponding author upon request.

## Code availability

The codes that support the findings of this study are available from the corresponding author upon request.

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

## Acknowledgements
This work is funded by National Natural Science Foundation of China (62071010 X.X.).

## Author contributions

P.L., M.X., Z.C., and X.X. jointly proposed the design of the anti-resonant acoustic waveguide. P.L. and M.X. fabricated the waveguide devices. P.L. developed the etching technique and optimization algorithm, performed the experiments and analyzed the results. P.L., M.X., Y.B., and X.X. wrote the manuscript. All co-authors contributed by providing their valuable feedback and comments. X.X. supervised the work.

## Competing interests
The authors declare no competing interests.
