## [Peer Review File · Nature Communications]

Anti-resonant acoustic waveguides enabled tailorable Brillouin scattering on chipReviewer #1 (Remarks to the Author):

In the paper entitled "Anti-resonant acoustic waveguides enabled on-chip adaptable Brillouin scattering", the authors use a method that has already (4 years ago) been theoretically published by another group in New Journal of Physics (MK. Schmidt et al 2020 New J. Phys. 22 053011) to better confine acoustic modes in optical waveguides and improve optoacoustic interactions like Brillouin scattering. The method of using anti-resonant acoustic waveguides is presented as the novelty of the article as part of the third line of the abstract. However, since the method was already published by other authors in 2020, the authors should deeply revise their claim about novelty. The paper itself has enough novelty that I recommend its publication in Nature Communications since the same concept is nicely demonstrated, both theoretically and experimentally, in suspended periodic photonic integrated waveguides. The paper is well written and the experimental results are quite convincing and potentially promising for Brillouin-based applications. I do, however, have several critical points that should be addressed and revised by the authors before publication.

1- The concept of phononic band gaps in optical waveguides was first introduced by V. Laude et al. (Phys. Rev. B 71, 045107 –2005). This article should be cited in the introduction when they say: "the third strategy uses phononic bandgaps to obstruct the propagation of acoustic waves."

2- Authors should remove any misleading claims as "record-breaking", or "breakthroughs". It's unnecessarily oversold and this is quite far from what can be achieved in standard optical fibers in terms of Brillouin gain and linewidth.

3- The authors are expected to show a direct experimental comparison between anti-resonant suspended acoustic waveguides and conventional suspended-core waveguides to clearly demonstrate the benefit of anti-resonant acoustic confinement.

4- All previous studies have clearly shown that the net Brillouin gain (forward or backward) in SOI platforms is very weak compared to silica fibers due to strong acoustic attenuation (See reference Below). This problem should be discussed in detail in the revised paper.

Flavien Gyger et al. Phys. Rev. Lett. 124, 013902, 2020.

Reviewer #2 (Remarks to the Author):

Overall, I find this submission to be a valuable contribution to the field, presenting a novel approach to enhancing Stimulated Brillouin Scattering (SBS) in silicon chips through the implementation of the ARROW waveguide concept. The experimental demonstration is a significant step forward in a field that has seen recent progress from various groups exploring different platforms and geometries.

The authors appropriately acknowledge the well-established concept of ARROW confinement in optical waveguides and its recent theoretical consideration for acoustic/SBS confinement. The novelty lies in this being the first experimental demonstration, which adds substantial weight to the study. The reported results align well with the modeling, and the observed improvement in SBS gain showcases the practicality and novelty of the ARROW design.

By designing the anti-resonant structures the authors demonstrate they can flexibly control the acoustic modes without affecting optical modes, which underlies the flexibility that is achieved, leading to enhancement.

The claim of achieving a record SBS gain for silicon is commendable, but I suggest the authors elaborate on the potential limitations posed by nonlinear loss. Additionally, the under-etching of the sample is noted as a limitation in comparison to other chip platforms. It would be beneficial for the authors to address whether they envision generalizing this approach, perhaps in 3D, to eliminate the need for under-etching, thus enhancing the robustness of the design.

Figure 3c seems asymmetric. Can the authors explain the shape?

The explanation of the results is generally clear, but I encourage the authors to provide more intuitive insights into why this type of confinement offers enhanced SBS relative to strip waveguides. This would aid readers in better grasping the significance of the proposed ARROW design. Specifically, what exactly is the attribute of the arrow design that enables increase in the SBS gain, relative, for example to Rakich's record result in silicon.

Furthermore, the manuscript could benefit from an expansion of the literature review, particularly in relation to ARROW waveguides. Consider incorporating relevant works such as N. M. Litchinitser et al.'s "Antiresonant reflecting photonic crystal optical waveguides" (Opt. Lett. 27, 1592–1594, 2002). Additionally, for a deeper understanding of acoustic confinement, I recommend including Christopher G. Poulton et al.'s work on "Acoustic confinement and stimulated Brillouin scattering in integrated optical waveguides" (J. Opt. Soc. Am. B 30, 2657-2664, 2013) as a reference.

Lastly, the authors are encouraged to provide insights on how to interface with these modes in a typical circuit, adding practical applicability to their findings. Overall, with these minor revisions, I believe this manuscript has the potential to make a valuable contribution to Nature Communications.

Reviewer #3 (Remarks to the Author):

This paper presents a novel waveguide structure based on the acoustic antiresonance in the silicon-on-insulator platform. This anti-resonant acoustic waveguide can achieve strong SBS effect in both the forward and backward processes. Compared to previous SOI waveguide structures for strong SBS effect, the advantages on the fabrication side in this paper are evident. This paper is clear written, and the novelty is clear, I think this paper meets the standard for publication in Nature Communications if the following concerns can be addressed:

1. The waveguide structure in the paper is still suspended. From Fig.2b, the Brillouin gain coefficient tends to increase with t_1 . If t_1 goes to infinite, the waveguide structure would become more like a normal suspended SOI waveguide, rather than SARAW. How much is the Brillouin gain enhancement is contributed by the suspended structure, and how much is contributed by the acoustic antiresonance?
2. Is it possible to realize the acoustic antiresonance condition in a non-suspended SOI waveguide? The authors should provide the comparison between the SBS responses of the suspended SARAW and non-suspended SARAW.
3. How can the acoustic antiresonance condition be simultaneously satisfied for both the forward and backward SBS process? It makes sense that optimizing the slot thickness can create the antiresonance condition for the phonon in the transverse direction, which is involved in the forward SBS process. However, for the backward SBS, the phonon is the longitudinal direction, how can the anti-resonance condition be satisfied as well? The author should provide more explanations about that.
4. In terms of the genetic algorithm optimization, the objective function is the Brillouin gain coefficient for the forward SBS process. How far is from the theoretical limit of the Brillouin gain coefficient for the SARAW? Moreover, is it possible to optimize the Brillouin gain coefficient for the backward SBS process?
5. What is the reproducibility of this structure? I strongly suggest the authors to provide insights and comments on the reproducibility and scalability of the SARAW on a wafer scale

Some minor concerns:

1. The usage of 'thickness' in the manuscript is confusing, it refers to the width of the slot.
2. The authors should provide a table that compares the Brillouin nonlinearities between this structure and other silicon waveguides in the literature.

We thank the reviewers for their thorough reading and valuable comments to improve the manuscript quality. We have correspondingly responded to all the comments. Please see the details of the point-by-point response to these comments below.

Response to Reviewer #1

General comment:

In the paper entitled “Anti-resonant acoustic waveguides enabled on-chip adaptable Brillouin scattering”, the authors use a method that has already (4 years ago) been theoretically published by another group in New Journal of Physics (MK. Schmidt et al 2020 New J. Phys. 22 053011) to better confine acoustic modes in optical waveguides and improve optoacoustic interactions like Brillouin scattering. The method of using anti-resonant acoustic waveguides is presented as the novelty of the article as part of the third line of the abstract. However, since the method was already published by other authors in 2020, the authors should deeply revise their claim about novelty. The paper itself has enough novelty that I recommend its publication in Nature Communications since the same concept is nicely demonstrated, both theoretically and experimentally, in suspended periodic photonic integrated waveguides. The paper is well written and the experimental results are quite convincing and potentially promising for Brillouin-based applications. I do, however, have several critical points that should be addressed and revised by the authors before publication.

Reply: We would like to express our gratitude to the reviewer for his/her thorough reading and positive feedback. For the article [R1] mentioned by the reviewer, we have had referenced and commented on this work in the introduction of the anti-resonant reflection concept from lines 62 to 72 in our manuscript. The theoretical framework of our study was indeed inspired by this article, and Supplementary Section III also cited this work. However, the structural design of SARAWs is different from that presented in Ref. [R1], as shown in Fig. R1. To enhance phonon confinement, we have employed a suspended waveguide design. Additionally, the implementation of anti-resonant reflecting layers deviates from Ref. [R1], which utilizes different material layers (silicon and silica). In our design, anti-resonant reflection is achieved through geometric softening at the etched slots. This design is compatible with CMOS technology, facilitating the integration of anti-resonant reflection on integrated platforms.

Fig. R1 a. Suspended anti-resonant acoustic waveguide. b. Anti-resonant reflecting acoustic waveguide [R1].

Action: We have revised our claim in the Abstract: “Here, inspired by the optical anti-resonance in hollow-core fibers and acoustic anti-resonance in cylindrical waveguides, we propose suspended anti-resonant acoustic waveguides (SARAWs) with superior confinement and high selectivity of acoustic modes, supporting both forward and backward SBS on chip.”

Reference:

[R1] Schmidt, M. K., O’Brien, M. C., Steel, M. J., & Poulton, C. G. (2020). ARRAW: Anti-resonant reflecting acoustic waveguides. New Journal of Physics, 22(5), 053011.

Comment 1: The concept of phononic band gaps in optical waveguides was first introduced by V. Laude et al. (Phys. Rev. B 71, 045107 –2005). This article should be cited in the introduction when they say: "the third strategy uses phononic bandgaps to obstruct the propagation of acoustic waves."

Reply: Thanks for the reviewer's comment. We have added this reference.

Action: We have added this reference in the second paragraph of the introduction: "The third strategy utilizes phononic bandgaps to obstruct the spread of acoustic waves [28–31]. However, its complex structures pose difficulties for design and manufacturing and result in low robustness."

Comment 2: Authors should remove any misleading claims as "record-breaking", or "breakthroughs". it's unnecessarily oversold and this is quite far from what can be achieved in standard optical fibers in terms of Brillouin gain and linewidth.

Reply: As pointed out by the reviewer, the current standard optical fibers with a length of 10 km can easily achieve Brillouin gain exceeding 50dB [R2]. However, this paper primarily emphasizes how the SARAW structure brings breakthroughs and enhancements to centimeters-long on-chip silicon waveguides, thereby extending this approach to other integrated material platforms. In fact, our use of words like "record-breaking" or "breakthrough" is explicitly limited, such as "on the SOI platform" or "in integrated silicon waveguides." To address the reviewer's comments and maintain the manuscript's rigor, we have made adjustments to some statements.

Action:

1. Abstract: "Leveraging the advantages of SARAWs, we have showcased a series of **breakthroughs** for SBS within a compact footprint on the silicon-on-insulator platform."
2. The fourth paragraph: "Consequently, SARAWs can support both forward and backward SBS and have achieved a series of **extraordinary results** of SBS on the silicon-on-insulator (SOI) platform."
3. Page five, second paragraph: "The 700-nm SARAW achieve a **remarkable** G_B of $3530 W^{-1}m^{-1}$ in silicon waveguides with a balance of G_B/Q_m and Q_m ."

Reference:

[R2] Eggleton, B. J., Poulton, C. G., Rakich, P. T., Steel, M. J., & Bahl, G. (2019). Brillouin integrated photonics. Nature Photonics, 13(10), 664-677.

Comment 3: The authors are expected to show a direct experimental comparison between anti-resonant suspended acoustic waveguides and conventional suspended-core waveguides to clearly demonstrate the benefit of anti-resonant acoustic confinement.

Reply: In our subsequent work [R3], we proposed a suspended nanowire structure with anti-resonant supporting arms, as illustrated in Fig. R2. This structure, derived from the SARAW (Fig. R1a), eliminates the flanked anti-resonant structures and retains the anti-resonant design only at the supporting arms. With the same waveguide width of 1200 nm, the suspended nanowire exhibits reductions in both Brillouin gain coefficient (G_B) and mechanical quality factor (Q_m) compared to SARAWs (Suspended nanowire: $1100 W^{-1}m^{-1}$, 550; SARAW: $1670 W^{-1}m^{-1}$, 650).

This can be attributed to the following reasons: First, in SARAWs, the anti-resonant reflection can compress the acoustic mode into the central waveguide, thereby mitigating the impact of sidewall roughness on the Q_m , resulting in a larger G_B . Second, the presence of the anti-resonant structure can isolate the optical mode from the supporting arms, reducing optical mode mismatch

losses. Third, in terms of fabrication, the SARAW is more amenable to manufacture and exhibits better structural stability.

Regarding the distinction between SARAWs and suspended rib waveguides, a similar question was proposed by Reviewer #2. Please refer to the response provided in Comment 4 from Reviewer #2.

Action: At the end of the first paragraph on page five, we have added a description: “Compared to suspended rectangular waveguides without anti-resonant structures, SARAWs can compress the acoustic mode into the central waveguide, thereby mitigating the impact of sidewall roughness on the Q_m , resulting in a larger G_B . Moreover, the presence of the anti-resonant structure can isolate the optical mode from the supporting tethers, reducing optical mode mismatch loss.”

Reference:

[R3] Peng, L., Mingyu, X., Yunhui, B., Zhangyuan, C., & Xiaopeng, X. (2024). Loading-effect-based 3-D microfabrication empowers on-chip Brillouin optomechanics. arXiv preprint arXiv:2402.02409.

Fig. R2 Suspended nanowire [R3].

Comment 4: All previous studies have clearly shown that the net Brillouin gain (forward or backward) in SOI platforms is very weak compared to silica fibers due to strong acoustic attenuation (See reference Below). This problem should be discussed in detail in the revised paper. Flavien Gyger et al. Phys. Rev. Lett. 124, 013902, 2020.

Reply: As discussed in comment 2, achieving larger Brillouin net gain is more attainable in standard optical fibers compared to the current capabilities of the SOI platform. However, we attribute this primarily to the optical losses, especially the nonlinear absorption losses of the SOI platform. In fact, for waveguide lengths in the centimeter range, integrated Brillouin waveguides can exhibit higher mechanical quality factor than optical fibers, indicating lower acoustic attenuation. The SOI platform holds distinct advantages in Brillouin gain coefficient and mechanical quality factor, which are beneficial for specific applications such as filters and on-chip Brillouin sensing. Additionally, introducing far-infrared wavelengths and free-carrier suppression techniques [R5] will contribute to enhancing the Brillouin net gain on the SOI platform. It is noteworthy that SARAWs can be extended to other material platforms. For instance, in previous work with chalcogenide glass [R4], a Brillouin net gain of 40 dB has already been achieved.

Action: We have added an additional segment at the end of the first paragraph on page six: “These results mark the present apex of Brillouin gain levels in silicon waveguides and can be further improved with higher on-chip power and longer waveguide length. However, it should be noted that, on the SOI platform, further increases in pump power may lead to notable nonlinear absorption losses (see Supplementary Section I). This factor would become the primary constraint for achieving larger Brillouin net gain. The introduction of pump light in far-infrared wavelengths or the free-carrier suppression techniques could potentially overcome this limitation.”

Reference:

[R4] Choudhary, A., Morrison, B., Aryanfar, I., Shahnia, S., Pagani, M., Liu, Y., ... & Eggleton, B. J. (2016). Advanced integrated microwave signal processing with giant on-chip Brillouin gain. *Journal of lightwave technology*, 35(4), 846-854.

[R5] Turner-Foster, A. C., Foster, M. A., Levy, J. S., Poitras, C. B., Salem, R., Gaeta, A. L., & Lipson, M. (2010). Ultrashort free-carrier lifetime in low-loss silicon nanowaveguides. *Optics express*, 18(4), 3582-3591.

Response to Reviewer #2

General comment:

Overall, I find this submission to be a valuable contribution to the field, presenting a novel approach to enhancing Stimulated Brillouin Scattering (SBS) in silicon chips through the implementation of the ARROW waveguide concept. The experimental demonstration is a significant step forward in a field that has seen recent progress from various groups exploring different platforms and geometries.

The authors appropriately acknowledge the well-established concept of ARROW confinement in optical waveguides and its recent theoretical consideration for acoustic/SBS confinement. The novelty lies in this being the first experimental demonstration, which adds substantial weight to the study. The reported results align well with the modeling, and the observed improvement in SBS gain showcases the practicality and novelty of the ARROW design.

By designing the anti-resonant structures, the authors demonstrate they can flexibly control the acoustic modes without affecting optical modes, which underlies the flexibility that is achieved, leading to enhancement.

Reply: We would like to express our gratitude to the reviewer for his/her thorough reading and positive feedback.

Comment 1: The claim of achieving a record SBS gain for silicon is commendable, but I suggest the authors elaborate on the potential limitations posed by nonlinear loss.

Reply: We appreciate the valuable suggestions from the reviewer. Indeed, the SOI platform faces limitations attributed to nonlinear losses, including two-photon absorption and free carrier absorption. Both of these losses have been detailed in Supplementary Section I. Additionally, the gain curves in Fig. 3b and 3d also take into account these losses. Due to the output power constraints of our EDFA, the on-chip pump power is capped at a maximum of 60 mW. At this point, the increase in Brillouin net gain outweighs the nonlinear absorption losses. However, as the pump power increases to 200 mW, the Brillouin net gain plateaus at 10 dB, with nonlinear losses becoming the primary limitation. Therefore, to achieve a larger net gain on the SOI platform, we suggest not only increasing the pump power but also considering lengthening the waveguide. Additionally, efforts to suppress both linear and nonlinear losses [R6] in the waveguide are crucial.

Action: We have added an additional segment at the end of the first paragraph on page six: “These results mark the present apex of Brillouin gain levels in silicon waveguides and can be further improved with higher on-chip power and longer waveguide length. **However, it should be noted that, on the SOI platform, further increases in pump power may lead to notable nonlinear absorption losses (see Supplementary Section I). This factor would become the primary constraint for achieving larger Brillouin net gain. The introduction of pump light in far-infrared wavelengths or the free-carrier suppression techniques could potentially overcome this limitation.**”

Reference:

[R6] Turner-Foster, A. C., Foster, M. A., Levy, J. S., Poitras, C. B., Salem, R., Gaeta, A. L., & Lipson, M. (2010). Ultrashort free-carrier lifetime in low-loss silicon nanowaveguides. *Optics express*, 18(4), 3582-3591.

Comment 2: Additionally, the under-etching of the sample is noted as a limitation in comparison

to other chip platforms. It would be beneficial for the authors to address whether they envision generalizing this approach, perhaps in 3D, to eliminate the need for under-etching, thus enhancing the robustness of the design.

Reply: For the SOI platform, the acoustic velocity of the silica substrate is significantly smaller than that of silicon. This results in a substantial energy leakage into the substrate when guiding acoustic waves in the silicon membrane. In this work, we performed wet-etching on the underlying silica substrate, implementing an anti-resonant design exclusively in the horizontal direction. However, we propose that growing silica-silicon layers vertically to establish anti-resonant reflecting layers could also be effective, as shown in Fig. R3. This design might eliminate the necessity for the under-etching step. Additionally, for materials like chalcogenide glass or aluminum nitride, which have acoustic velocities smaller than the silica substrate, acoustic waves are unable to leak downward. The SARAWs structure can be seamlessly transplanted onto these platforms.

Fig. R3 Anti-resonant reflection in the vertical direction.

Action: We have added an image and a paragraph in Supplementary Section IV: “**It should be noted that, for SARAWs, to prevent acoustic wave leakage into the silica substrate, a 10% HF under-etching step is employed to achieve waveguide suspension. It adopts an anti-resonant design only in the horizontal direction. However, we propose that growing silica-silicon layers vertically to establish anti-resonant reflecting layers could also be effective, as illustrated in FigS.5. This design might eliminate the need for the under-etching step. Additionally, for materials like chalcogenide glass or aluminum nitride, which have acoustic velocities smaller than the silica substrate, acoustic waves are unable to leak downward. The SARAW structure can be seamlessly transplanted onto these platforms.**”

Comment 3: Figure 3c seems asymmetric. Can the authors explain the shape?

Reply: Fig. 3c is the result of the heterodyne FWM experiment, serving as a reference for the three-tone direct gain experiment. Similar to Fig. 2f, it exhibits a Fano-like line shape. For detailed information, please refer to Supplementary Section V.

Action: We have made the following modifications in the second paragraph on page five: “We extract Brillouin frequency shift, G_B , and Q_m by fitting the obtained **asymmetric** Fano-like spectral lines (see Supplementary Section V).”

Comment 4: The explanation of the results is generally clear, but I encourage the authors to provide more intuitive insights into why this type of confinement offers enhanced SBS relative to strip waveguides. This would aid readers in better grasping the significance of the proposed ARROW design. Specifically, what exactly is the attribute of the arrow design that enables increase in the SBS gain, relative, for example to Rakich's record result in silicon.

Reply: Regarding the distinction between SARAWs and strip/rectangular waveguides, a similar question was proposed by Reviewer #1. Please refer to the response provided in Comment 3 from Reviewer #1.

In comparison to the suspended rib waveguides proposed by Rakich research group [R7], SARAWs have the following advantages. First, as shown in Fig. R4a, the acoustic mode field of rib waveguides is also distributed on both sides of the rib. In contrast, SARAWs squeeze the acoustic mode field into the central waveguide, resulting in a larger photon-phonon overlap. Second, compared to rib waveguides, SARAWs only require a single ICP etching step, which is beneficial for suppressing the inhomogeneous broadening of Brillouin resonance (see Supplementary Section VIII).

Action: We have made the following modifications to the third paragraph on page three: “The acoustic mode of SARAWs, **compared to the rib waveguides [22]**, is squeezed towards the central waveguide, as shown in Fig.1f, indicating the effectiveness of anti-resonant reflection in acoustic mode confinement.”

Reference:

[R7] Kittlaus, E. A., Shin, H., & Rakich, P. T. (2016). Large Brillouin amplification in silicon. *Nature Photonics*, 10(7), 463-467.

Fig. R4 a. The elastic displacement field of the acoustic mode of suspended rib waveguides [R7]. b. The elastic displacement field of the acoustic mode of SARAWs.

Comment 5: Furthermore, the manuscript could benefit from an expansion of the literature review, particularly in relation to ARROW waveguides. Consider incorporating relevant works such as N. M. Litchinitser et al.'s "Antiresonant reflecting photonic crystal optical waveguides" (*Opt. Lett.* 27, 1592–1594, 2002). Additionally, for a deeper understanding of acoustic confinement, I recommend including Christopher G. Poulton et al.'s work on "Acoustic confinement and stimulated Brillouin scattering in integrated optical waveguides" (*J. Opt. Soc. Am. B* 30, 2657-2664, 2013) as a reference.

Action: Thanks for the reviewer’s suggestions. We have added these references into the third and first paragraphs of the introduction respectively.

“Recently, integrated photonics has enabled better confinement and manipulation of optical and acoustic modes [19–27], ushering in a new era of photon-phonon interaction.”

“Emerging anti-resonant reflection offers an alternative approach to achieve field confinement [32–36]”

Comment 6: Lastly, the authors are encouraged to provide insights on how to interface with these modes in a typical circuit, adding practical applicability to their findings.

Reply: As detailed in Supplementary Section IV, our fabrication process is well-compatible with CMOS technology, facilitating the large-scale integration of Brillouin-based devices. Additionally, as discussed in the response of Comment 2, with an ingenious design, SARAWs may have the potential to eliminate the under-etching step. It not only further streamlines the fabrication process but also enhances its adaptability to various integrated photonic devices. Moreover, since the loading effect

is a common occurrence during the etching step of various materials, SARAWs can be transplanted onto other integrated photonic circuits.

Action: We have added an image and a paragraph in Supplementary Section IV: “**It should be noted that, for SARAWs, to prevent acoustic wave leakage into the silica substrate, a 10% HF under-etching step is employed to achieve waveguide suspension. It adopts an anti-resonant design only in the horizontal direction. However, we propose that growing silica-silicon layers vertically to establish anti-resonant reflecting layers could also be effective, as illustrated in FigS.5. This design might eliminate the need for the under-etching step. Additionally, for materials like chalcogenide glass or aluminum nitride, which have acoustic velocities smaller than the silica substrate, acoustic waves are unable to leak downward. The SARAW structure can be seamlessly transplanted onto these platforms.**”

Comment 6: Overall, with these minor revisions, I believe this manuscript has the potential to make a valuable contribution to Nature Communications.

Reply: We thank the reviewer for his/her positive support.

Response to Reviewer #3

General comment:

This paper presents a novel waveguide structure based on the acoustic antiresonance in the silicon-on-insulator platform. This anti-resonant acoustic waveguide can achieve strong SBS effect in both the forward and backward processes. Compared to previous SOI waveguide structures for strong SBS effect, the advantages on the fabrication side in this paper are evident. This paper is clear written, and the novelty is clear, I think this paper meets the standard for publication in Nature Communications if the following concerns can be addressed:

Reply: We would like to express our gratitude to the reviewer for his/her thorough reading and positive feedback.

Comment 1: The waveguide structure in the paper is still suspended. From Fig.2b, the Brillouin gain coefficient tends to increase with t_1 . If t_1 goes to infinite, the waveguide structure would become more like a normal suspended SOI waveguide, rather than SARAW. How much is the Brillouin gain enhancement is contributed by the suspended structure, and how much is contributed by the acoustic antiresonance?

Reply: In fact, the Brillouin gain coefficient will not continually increase. In Fig. 2b and Fig. 2d, the rising trend of G_B/Q_m is induced by the anti-resonance condition of another acoustic mode in the etched slots. As t_1 further increases, the resonance condition of this acoustic mode results in a reduction in acoustic energy density and G_B/Q_m , as illustrated in FigS. 3b in Supplementary Section III.

Regarding the differences between SARAWs compared to suspended strip/rectangular waveguides and suspended rib waveguides, a similar question was proposed by Reviewer #2. Please refer to the response provided in Comment 4 from Reviewer #2.

Additionally, it should be noted that t_1 cannot increase indefinitely. Even when operating under anti-resonance conditions, the anti-resonant reflectance of the etched slots falls short of reaching 100%, leading to residual acoustic energy distribution within these slots. As t_1 increases, the energy distribution within the etched slots grows. This results in a decrease in the photon-phonon overlap. Furthermore, an excessive rise in t_1 is associated with a deterioration in structural stability, rendering the waveguide more prone to collapse.

Lastly, we discuss the contributions of suspension and anti-resonance to SARAWs. For the SOI platform, the absence of suspension design would lead to significant acoustic leakage into the silica substrate, posing a significant challenge in detecting SBS. Moreover, as discussed in our response in Comment 3 from Reviewer #1 and detailed in our subsequent work [R3], the removal of the anti-resonant structures resulted in a decrease in the Brillouin gain coefficient from 1670 to 1100 $W^{-1}m^{-1}$ and a decline in the mechanical quality factor from 650 to 550.

Action:

1. At the end of the first paragraph on page five, we have added a description: **“Compared to suspended rectangular waveguides without anti-resonant structures, SARAWs can compress the acoustic mode into the central waveguide, thereby mitigating the impact of sidewall roughness on the Q_m , resulting in a larger G_B . Moreover, the presence of the anti-resonant structure can isolate the optical mode from the supporting tethers, reducing optical mode mismatch loss.”**
2. We have made the following modifications to the third paragraph on page three: **“The acoustic mode of SARAWs, compared to the rib waveguides [22], is squeezed towards the central**

waveguide, as shown in Fig.1f, indicating the effectiveness of anti-resonant reflection in acoustic mode confinement.”

Comment 2: Is it possible to realize the acoustic antiresonance condition in a non-suspended SOI waveguide? The authors should provide the comparison between the SBS responses of the suspended SARAW and non-suspended SARAW.

Reply: Please refer to the response in Comment 2 from Reviewer #2. With an ingenious design, we believe that the implementation of a non-suspended anti-resonant SOI waveguide is achievable. However, in this paper, without suspension, the acoustic waves would essentially leak into the silica substrate, making the observation of SBS challenging.

Action: We have added an image and a paragraph in Supplementary Section IV: “**It should be noted that, for SARAWs, to prevent acoustic wave leakage into the silica substrate, a 10% HF under-etching step is employed to achieve waveguide suspension. It adopts an anti-resonant design only in the horizontal direction. However, we propose that growing silica-silicon layers vertically to establish anti-resonant reflecting layers could also be effective, as illustrated in FigS.5. This design might eliminate the need for the under-etching step. Additionally, for materials like chalcogenide glass or aluminum nitride, which have acoustic velocities smaller than the silica substrate, acoustic waves are unable to leak downward. The SARAW structure can be seamlessly transplanted onto these platforms.**”

Comment 3: How can the acoustic antiresonance condition be simultaneously satisfied for both the forward and backward SBS process? It makes sense that optimizing the slot thickness can create the antiresonance condition for the phonon in the transverse direction, which is involved in the forward SBS process. However, for the backward SBS, the phonon is the longitudinal direction, how can the anti-resonance condition be satisfied as well? The author should provide more explanations about that.

Reply: First of all, it should be noted that the acoustic modes for forward SBS and backward SBS are distinct. This leads to differences in their anti-resonance conditions and geometric parameters, as shown in Fig. 2f and Fig. 4c. Separate genetic algorithm optimizations have been performed for forward and backward SBS.

The theoretical exposition on how the anti-resonance conditions are satisfied by the acoustic modes of backward SBS is detailed in the Ref. [35], as cited in the third paragraph of the Introduction. The cylindrical waveguide proposed in this reference facilitates backward SBS. Essentially, this arises from the fact that, even though the main component of the backward SBS acoustic wave is a longitudinal, its wave-vector still possesses a component in the transverse direction, leading to the corresponding anti-resonance condition, as shown in Fig. R5.

Action:

1. We have modified a sentence in the third paragraph of the Introduction: “Anti-resonant reflection has been theoretically proposed to confine acoustic modes of **both forward and backward SBS** in cylindrical waveguides.”
2. We have added a paragraph in Supplementary Section VII: “**SARAWs also have the capability to support backward Brillouin scattering. This arises from the fact that, even though the main component of the backward SBS acoustic wave is longitudinal, its wave-vector still possesses a**

component in the transverse direction, leading to the corresponding anti-resonance condition.”

Fig. R5. The propagation of the acoustic waves along the cylindrical waveguide is characterized by the wavenumber β , which is constant throughout the structure, and the transverse wavenumber $k_{s/p}$ for longitudinal (P) and shear (S) waves [R1].

Comment 4: In terms of the genetic algorithm optimization, the objective function is the Brillouin gain coefficient for the forward SBS process. How far is from the theoretical limit of the Brillouin gain coefficient for the SARAW? Moreover, is it possible to optimize the Brillouin gain coefficient for the backward SBS process?

Reply: Firstly, investigating the Brillouin gain coefficient of SARAWs involves solving a problem with complex boundary conditions in multiple physical fields, and this problem lacks an analytical solution. We can only approach the optimal solution and theoretical limit according to the algorithmic convergence. The coupling factor G_B/Q_m obtained from our simulation exhibits great consistency with experimental results, as illustrated in Fig. 2d.

Additionally, as mentioned in the response of Comment 3, the anti-resonance condition and geometric parameters for backward SBS are also obtained through genetic algorithm optimization, ensuring the optimal backward Brillouin gain coefficient.

Action: Please refer to the action in the previous comment.

Comment 5: What is the reproducibility of this structure? I strongly suggest the authors to provide insights and comments on the reproducibility and scalability of the SARAW on a wafer scale.

Reply: This structure exhibits great reproducibility and scalability. As detailed in Supplementary Section IV, the loading effect of the etching device can be finely adjusted and precisely measured. Utilizing the obtained relationship between slot width and etching depth, genetic algorithm optimization can be accurately performed. Moreover, during our experiments, benefiting from the robust structural support and compact footprint, the devices exhibit excellent stability and a high yield rate on a wafer scale. This enables the convenient replication of the SARAW structure on different material platforms and fabrication equipment, promoting the exploration and integration of Brillouin-based devices.

Action: We have added a sentence at the end of the fourth paragraph of the third page: “Benefiting from the streamlined fabrication process and compatibility with CMOS technology, these results exhibit great reproducibility and scalability, facilitating the large-scale integration of Brillouin-based devices.”

Comment 6: The usage of 'thickness' in the manuscript is confusing, it refers to the width of the slot.

Reply: The use of "thickness" continues the previously used phrase "thickness of anti-resonant reflecting layers." In response to the reviewer's comment and to avoid confusion, we have revised "thickness of the slot" to "width of the slot." However, we have retained the symbol t_i in the description, as the symbol w_i associated with width has been used.

Action: We have revised "thickness of the slot" to "width of the slot."

Comment 7: The authors should provide a table that compares the Brillouin nonlinearities between this structure and other silicon waveguides in the literature.

Reply: SBS encompasses several key metrics, including the mechanical quality factor, shift frequency, gain coefficient, and net gain. Assessing these metrics requires a comprehensive comparison of parameters like optical loss, waveguide length, and pump power. To maintain the coherence and focus of the main text, we have added a table in Supplementary Section VII to contrast the Brillouin nonlinearities of silicon waveguides in recent works.

Action: We have added a table and a paragraph in Supplementary Section VII: "Referring to RefS.[17], we further compared our results of backward and forward SBS in SARAWs with recent studies conducted on silicon waveguides [1,12,14-16]. As shown in Tables.1, SARAWs have not only lower optical losses but also significantly higher mechanical quality factors and Brillouin gain coefficients, supporting a net gain of up to 6.4 dB. Moreover, we achieved the largest Brillouin frequency and mechanical quality factor to date through backward SBS in SARAWs. The design of SARAWs allows flexible adjustments for these metrics. We believe that the structural and design methodology outlined in this paper has the potential to usher in comprehensive breakthroughs for integrated Brillouin waveguides."

TableS. 1. Comparison of Brillouin nonlinearity in integrated silicon waveguides.

Ref	Year	Length (cm)	Optical loss (dB/cm)	Scattering process	Brillouin frequency (GHz)	Mechanical quality factor	Gain coefficient ($W^{-1}m^{-1}$)	Net gain (dB)
[14]	2013	0.33	7	FSBS	1.8-16.3	1,000-1,800	2,750	-1.9
[15]	2015	4	0.18	FSBS BSBS	9.2 13.7	306 971	3,218 357	-0.1 -
[16]	2015	2.5	5.5	FSBS	9.1	728	6,561	0.5
[1]	2016	2.9	0.18	FSBS	4.4	680	1,152	5.2
[12]	2021	1.1	2	FSBS	4.45	566	1054	-1
This work	2024	2.5	0.3	FSBS BSBS	3.6-8.6 18.1-27.6	650-850 1,200-1,960	1,600-3,530 430-600	6.4 -

Reviewer #1 (Remarks to the Author):

I have read and carefully analyzed the authors' responses to all the referees' criticism. The revised manuscript addresses most of my initial concerns and the drawn conclusions are now supported by more data and discussions. I must say however that the lack of backward SBS net gain evidence weakens the paper. Nevertheless, I believe the authors did a good job of addressing most of my comments and revising their paper. I believe that despite the mentioned weakness in my first review, this paper can be published in Nature Communications as it may represent a landmark to foster the investigation of anti-resonant acoustic waveguides in other different photonic platforms, such as photonic crystal fibers and integrated optical devices.

Reviewer #2 (Remarks to the Author):

The authors have addressed my comments so i am satisfied and have nothing further to add. Nice work.

Reviewer #3 (Remarks to the Author):

The authors have addressed the concerns I raised in my previous review. In principle, I'm happy to recommend publication of this manuscript.

However, recently there is a paper published in Optics Letters (<https://doi.org/10.1364/OL.519929>) from the the same group focusing on the fabrication of very similar structures as reported in this manuscript.

In a first glance, wiht the structuring of the data presentation and the figures, these two manuscripts seem to have very large overlap. Upon more detailed inspection, one can appreciate subtle differences of the works.

But I'm still concerned about the large overlap in the fabricaiton sections of the two papers and I suggest modification of this manuscript to avoid an impression of publishing the same results twice. I'm happy to leace this decision to the editor.

We thank the reviewers again for their thorough reading and valuable comments to improve the manuscript quality. We have correspondingly responded to all the comments. Please see the details of the point-by-point response to these comments below.

Response to Reviewer #1

Comment1: I have read and carefully analyzed the authors' responses to all the referees' criticism. The revised manuscript addresses most of my initial concerns and the drawn conclusions are now supported by more data and discussions.

Reply: We express our gratitude to the reviewer for his/her invaluable support.

Comment2: I must say however that the lack of backward SBS net gain evidence weakens the paper.

Reply: As elucidated in the second paragraph on page five, the absence of net gain in backward SBS can be attributed to the limitations imposed by the material property of the photoelastic coefficient in silicon. However, despite these limitations, the SARAW structure achieves notable enhancements in the backward SBS gain coefficient, frequency shift, and quality factor within silicon waveguides. It underscores the superior characteristics of SARAWs in on-chip acousto-optic interactions. In pursuit of backward SBS net gain, two potential approaches may be considered:

1. Transplant SARAW onto a more favorable platform for backward SBS, such as chalcogenide glass.

2. Select SOI wafers with different crystal orientations to align larger photoelastic coefficients with the propagation direction of the waveguide [1].

Action: In the second paragraph of the page seven, we incorporated the following sentences: "As a result, SARAWs support acoustic modes with high group velocity ($> 3,000$ m/s), and the corresponding acoustic decay length of over $50 \mu\text{m}$, offering a low-loss and long-distance transmission platform for acoustic waves. **However, the observation of backward SBS net gain remains hindered by the limitation imposed by the intrinsic photoelastic coefficient of silicon. Promising avenues for overcoming this challenge include transplanting SARAWs onto a chalcogenide glass platform or selecting SOI wafers with diverse crystal orientations.**"

Reference: [1]. Diallo, S., Aubry, J. P., & Chembo, Y. K. (2017). Effect of crystalline family and orientation on stimulated Brillouin scattering in whispering-gallery mode resonators. *Optics Express*, 25(24), 29934-29945.

Comment3: Nevertheless, I believe the authors did a good job of addressing most of my comments and revising their paper. I believe that despite the mentioned weakness in my first review, this paper can be published in Nature Communications as it may represent a landmark to foster the investigation of anti-resonant acoustic waveguides in other different photonic platforms, such as photonic crystal fibers and integrated optical devices.

Reply: We thank the reviewer once again for his/her highly positive evaluation of our work. We also acknowledge the reviewer's insightful comments on the potential impact of our research. We do hope that the publication of this work will foster the investigation of anti-resonant acoustic waveguides in other different photonic platforms.

Response to Reviewer #2

General comment:

The authors have addressed my comments so i am satisfied and have nothing further to add. Nice work.

Reply: We extend our sincere gratitude to the reviewers for his/her support of our work.

Response to Reviewer #3

Comment1: The authors have addressed the concerns I raised in my previous review. In principle, I'm happy to recommend publication of this manuscript.

Reply: We would like to express our gratitude to the reviewer for his/her positive feedback.

Comment2: However, recently there is a paper published in Optics Letters (<https://doi.org/10.1364/OL.519929>) from the same group focusing on the fabrication of very similar structures as reported in this manuscript.

In a first glance, with the structuring of the data presentation and the figures, these two manuscripts seem to have very large overlap. Upon more detailed inspection, one can appreciate subtle differences of the works.

Reply: Actually, these two structures (as shown in this NC manuscript and published in Optics Letters) are different, each originating from distinct motivations. As illustrated in the Fig. R1, the anti-resonant structure in SARAWs plays a crucial role in tuning and controlling the acoustic mode characteristics of the waveguide. However, in the suspended nanowire structure (Optics Letters), we removed the anti-resonant structure to pursue further structural simplification. Notably, the SARAW showcased in this NC manuscript offers palpable advantages in terms of flexibility, backward SBS, mechanical quality factor, and Brillouin gain coefficient compared to the one shown in Optics Letters. Moreover, the new concept and context presented in this under-revision manuscript are significantly more comprehensive than those covered in the Optics Letters paper.

Additionally, the fabrication method in Optics Letters is further developed based on this under-revision manuscript. Therefore, we adopted a similar explanatory approach in the concept

Fig. R1 Comparison between (a) anti-resonant acoustic waveguides (under-revision manuscript) and (b) suspended nanowires (Optics Letters)

of the loading effect. However, to comply with copyright regulations, all figures underwent meticulous redrawn.

Action:

1. In the second paragraph of the fifth page, we added a sentence: “Moreover, the presence of the anti-resonant structure can isolate the optical mode from the supporting tethers, reducing optical mode mismatch loss. **These attributes underscore the comprehensive superiority and inherent flexibility of SARAWs.**”
2. We have further revised FigS.4 in Supplementary Section IV.

Comment3: But I'm still concerned about the large overlap in the fabrication sections of the two papers and I suggest modification of this manuscript to avoid an impression of publishing the same results twice. I'm happy to leave this decision to the editor.

Reply: Thank you for pointing out the potential misunderstanding regarding the fabrication approach. The fabrication approach presented in the Optics Letters paper was developed based on the ongoing revision of this NC manuscript, and should not be mistaken as submitting the same results to different journals.

On January 8, 2024, we submitted our manuscript to Nature Communication.

On January 23, 2024, the initial version of the NC manuscript was made available on arXiv (<https://doi.org/10.48550/arXiv.2401.12677>). Subsequently, on January 25, 2024, the Optics Letters paper was submitted, **referencing and citing the arXiv preprint to acknowledge the prior work**. The loading-effect-based fabrication method was initially developed in this under-revision manuscript. The Optics Letters paper then built upon this innovation and introduced a new type of suspended nanowire structure. As depicted in Fig. R2, distinct differences are apparent in the realized fabrication outcomes of the two articles.

On January 30, 2024, feedback from the first round of review of the NC manuscript was received. Some of the context presented in the Optics Letters paper addressed the questions raised by the NC reviewers. The initial version of the Optics Letters manuscript was posted on arXiv on February 4, 2024 (<https://doi.org/10.48550/arXiv.2402.02409>). The Optics Letters paper was accepted on February 20, 2024, and it was published online on March 5, 2024.

In response to the reviewer's suggestion, the fabrication figure in the supplementary materials was modified to avoid any misunderstanding or overlap.

Action:

1. We have further revised FigS.4 in Supplementary Section IV.
2. In the fourth paragraph of the third page of our manuscript, we further underscored the originality and novelty of this study: “Here, **for the first time**, we innovatively propose an etching technique based on the loading effect to fabricate the whole structure via a single exposure and etching step (see Methods and Supplementary Section IV).”

Fig. R2 Comparison of fabrication process between (a) anti-resonant acoustic waveguides (under-revision manuscript) and (b) suspended nanowires (Optics Letters)